# Characterization of subtypes and transmitted drug resistance strains of HIV among Beijing residents between 2001-2016

Jingrong Ye[1], Mingqiang Hao[1], Hui Xing[2], Yuncong Wang[1], Juan Wang[1], Yi Feng[2], Ruolei Xin[1], Ji Zeng[1], Shuai Zhao[2], Yinxiao Hao[1], Jing Chen[1], Yuhua Ruan[2], Xue Li[1], Yiming Shao[2]* , Hongyan Lu[1]* 

**1** Institute for HIV/AIDS and STD Prevention and Control, Beijing Center for Disease Prevention and Control and Beijing Research Center for Preventive Medicine, Beijing, China, **2** Division of Virology and Immunology, State KeyLaboratory for InfectiousDisease Prevention and Controland National Center for AIDS/STD Preventionand Control(NCAIDS), China Center for Disease Prevention and Control, Beijing, China

☯ These authors contributed equally to this work.
* hongyan_lu@sina.com (HL); yshao@bjmu.edu.cn (YS)

**Data Availability Statement:** The GenBank accession numbers of nucleotide sequences reported in this article are HQ007312–HQ007350, JF759957–JF760203,JF906562–JF906700,

## Abstract

### Background

Beijing is a national and international hub potentially containing a broad diversity of HIV variants. Previous studies on molecular epidemiology of HIV in Beijing pooled together samples from residents and non-residents. Pooling residents and non-residents has potentially introduced bias and undermined a good assessment and the intervention among the autochthonous population. Here, we aimed to define HIV subtype diversity and investigate the TDR in Beijing residents exclusively.

### Methods

We analyzed the demographic, clinical, and virological data collected between 2001 and 2016 from residents in Beijing. A population-based sequencing of the HIV *pol* gene was carried out using plasma specimens. Phylogenetic analysis was performed in order to classify sequences into their corresponding subtypes using an automated subtyping tool, the Context-Based Modeling for Expeditious Typing (COMET). Furthermore, the drug resistance mutations were determined using the World Health Organization list for surveillance of TDR mutations.

### Results

Data on TDR were available for 92% of 2,315 individuals with HIV infection, of whom 7.1% were women. The bioinformatic analysis of HIV strains from this study revealed that a combined 17 subtypes were circulating in Beijing, China between 2001 and 2016. The most common ones were CRF01_AE, CRF07_BC, and subtype B in Beijing during this period. The overall prevalence of TDR was 4.5% (95% confidence intervals[CI]: 3.6%–5.4%), with a declining trend over the period of spanning 2001 through 2016. In-depth

JF759957–JF760203, KM011653–KM011849, KY713346-KY713582.

**Funding:** This work was supported by Beijing Natural Science Foundation (D161100000416002 to Hongyan Lu and 7202074 to Ruolei Xin). The funder had no role in study design, data collection and analysis, decision to publish, or preparation of the manuscript.

**Competing interests:** The authors have declared that no competing interests exist.

class-specific analysis revealed that the prevalence of TDR for the nucleoside reverse-transcriptase inhibitors (NRTIs) was 1.0% (95% CI: 0.6–1.5), 0.9% (95% CI:0.6–1.4) for non-NRTIs and 2.8% (95% CI:2.1–3.5) for protease inhibitors. The prevalence of TDR was lower in individuals infected with the CRF07_BC HIV strain than those infected with CRF01_AE.

## Conclusions

Our data showed that the HIV epidemic in Beijing displayed a high genetic heterogeneity and a low and declining prevalence of TDR. In sharp contrast to Europe and North America, the declining trend of TDR between 2001 through 2016 was noticed while there was a wide-spread distribution of antiretroviral treatment in Beijing, China.

## Introduction

HIV epidemic in Beijing, the capital of the largest developing country, has remained stable over the last 5 years. By Oct 31, 2016, the Beijing HIV epidemiology database cumulatively recorded 21,886 HIV-positive individuals since the identification of the first case of AIDS in 1985 in China. Most of these cases are among the so called, the floating population also known as people without the Beijing Hukou identification (74.2%) and men who have sex with men (MSM, [66.0%]). In 2003, Beijing launched a vigorous intervention campaign as part of the China National Free Antiretroviral Treatment Program (NFATP). The NFATP had markedly improved the prognosis of individuals with HIV in Beijing. By the end of 2016, 13,221 individuals have been treated with antiretroviral drugs through NFATP[1,2]. However, there has been a general concern that the prevalence of transmitted drug resistance (TDR) could increase in parallel with the increasing availability of antiretroviral treatment (ART). Incidentally, such increase of TDR could negatively compromise the effectiveness of ART distribution program [3]. This concern is particularly important because in 2016, China implemented the World Health Organization (WHO) "treat-all", "treat-early" and "treatment as the prevention" policy [4,5]. Previous epidemiological studies documented a relatively high genetic diversity and prevalence of TDR of HIV in Beijing[6–9]. However, data in those studies were collected from both the non-resident floating population and the Beijing residents (people with Beijing Hukou). Indeed, these studies lacked adequate stratification for origin of subjects, and very little molecular information was available for the residents and the floating population. Continued monitoring the trend of TDR in a specific population can provide important insights that may inform clinical practice indicating which first-line ART regimens should be used. The analysis of the *pol* region can serve double purpose:1) the detection of TDR and 2) for subtype determination and phylogenetic analysis. The latter, can give insight into patterns of HIV transmission, with direct implications for public health policy[10]. In this study, we aimed to characterize the trend of the HIV subtype diversity and the prevalence of TDR in Beijing residents from 2001 to 2016.

## Materials and methods

### Ethics

The Research Ethics Committee in Beijing Center for Disease Prevention and Control(CDC) approved the study. By law, consent was not required because these data were collected and analyzed in the course of routine public health surveillance.

## Study patients

The Beijing HIV laboratory network (BHLN) was established in 1986 by the Beijing Municipal Commission of Health as a collaborative network of laboratories tasked to perform HIV diagnostic testing in Beijing. The BHLN includes a central HIV confirmatory laboratory in the Beijing CDC, four additional HIV confirmatory laboratories (DiTan, YouAn, Peking Union Medical College, and PLA General Hospital), and 280 HIV screening laboratories. The collaboration maintains a biobank with more than 50,000 stored samples collected from 21,886 individuals tested for HIV infection in Beijing since 1986. BHLN also maintains an HIV epidemiology database, which tracks patient diagnosed with HIV in Beijing and keeps records of the baseline of CD4 counts. BD FACS Calibur, BD FACS Canto II, and Beckman Coulter FC500 were used for CD4 cell counting. TDR was monitored in Beijing every year since 2006[6]. This involved a yearly survey of TDR among individuals newly diagnosed with HIV. A simple sampling scheme was designed to ensure broad representation of and feasibility of the survey. Briefly, samples were randomly selected from every other patient that was newly diagnosed with HIV infection. In addition, we included equal number of stored samples before the introduction of routine genotyping in Beijing, China in 2005. Inclusion criteria included, (1) being 18 years old or older, (2) being newly diagnosed with HIV and (3) not being pregnant. Individuals who reported previous use of antiretroviral drugs for treatment or prophylaxis were excluded from the present study.

## HIV subtyping

HIV subtype was inferred by automated subtyping using Context-Based Modeling for Expeditious Typing (COMET)-HIV[11]. Sequences classified as "unassigned" by COMET were further analyzed using neighbor-joining phylogenetic analysis. The phylogenetic trees were constructed using the Kimura 2-parameter model, with 1,000 bootstrap replicates, using the Mega 6.0 software.

## HIV TDR analyses

A population-based Sanger sequencing of the HIV protease gene and the deduced amino acid sequence from codon 1 through 300 of the reverse transcriptase gene of all specimens were analyzed using in-house methods[6,12]. All virological testing were performed at two reference laboratories: the Division of Research on Virology and Immunology, China CDC (for the 2011 and 2013 survey) and the Beijing Central HIV confirmatory laboratory, Beijing CDC (for the survey of the other years). Both laboratories participated in external quality assessment schemes for genotypic TDR testing from the National AIDS Reference Laboratory of the National Center for AIDS/STD Prevention and Control. Three commercial sequencing companies(Beijing Sino Geno Max Co., Ltd, Beijing Tsingke Biological technology Co., Ltd, and Beijing TianyiHuiyuan Biological technology Co., Ltd) performed the sequencing using the ABI 3500 Analyzer. These companies provided the external quality assessment for sequencing performed by our research team.

The TDR was determined in two steps. Firstly, the prevalence of TDR was determined using the Stanford Calibrated Population Resistance (CPR) method, based on the 2009 WHO list of surveillance of TDR mutation(STDRM)[13]. Secondly, for patients harboring a virus with at least one TDR mutation, the Stanford drug-susceptibility algorithm (version 8.5) was used to classify sequences as susceptible (Stanford level 1 or 2), low-level resistance (Stanford level 3), intermediate-level resistance (Stanford level 4), or high-level resistance (Stanford level 5) to the drug classes (nucleoside reverse transcriptase inhibitors [NRTIs], non-NRTIs [NNRTIs], and protease inhibitors [PIs]) and specific drugs.

## Data analysis

Baseline demographic data, transmission risk and CD4 cell counts were extracted from the Beijing HIV epidemiology database, ascertaining that patients information were anonymized and de-identified prior to analysis. Patients were grouped according to their residential status whether they hold the Beijing Hukou status (residents) or not (floating population). The Hukou system is a basic system of household registration in China. It officially identifies a person as a resident of an area. The Hukou includes identifying information such as name, parents, spouse, and date of birth. An individual without Hukou is regarded as an illegal resident.

The sampling time was divided into four phases: 2001–2008, 2009–2011, 2012–2014, and 2015–2016. Categorical and continuous data were compared using the $\chi^2$ test and with one-way ANOVA, respectively. The prevalence of TDR mutation was calculated and sequences containing at least one TDR mutation were further characterized as NRTIs, NNRTIs, and PIs. The risk factors for acquiring TDR mutations were estimated using logistic regression. The variables used for data analysis were sex, age (18–24, 25–44, 45–64, and ≥65 years), ethnicity, HIV subtype, CD4 cell counts (<200, 200–349, 350–499, and ≥500 cells per μL), transmission risk group, and sampling phase. In the model, we included a binary response, indicating detection of any TDR mutation from each patient as an outcome.

We analyzed variables independently and included those that were associated ($p < 0.1$) with the outcome in the multivariable model. The results were expressed as odds ratios (ORs) with 95% confidence intervals (CIs) and two-sided P values, with a P value of <0.05 considered statistically significant. All analyses were performed using R (version 3.6.1)[14]. We used listwise deletion approach to handle missing data throughout the study. However, since 12.8% of data were missing for CD4 count, a sensitivity analysis was performed using multiple imputation to handle missing data (m = 5).

## Results

### Study population

The Beijing HIV epidemiology database keeps records of new cases of HIV diagnosed among Beijing residents. From 2001 to 2016, 4,784 new cases of HIV recorded in the Beijing database, of which half (n = 2,350) were selected for the purpose of the current study. Thirty-five individuals were excluded from the final analysis (n = 2,315) for being younger than 18 years old. Of 2,315 participants, genotype information was available for 2,130 (92.0%). Specimens without genotype occurred at random and the prevalence was within the expected range (S1 Table). To ensure that exclusion of patients did not introduce a bias in the data analysis, patients information of the excluded population were compared with the remaining study group. Indeed, the demographic data, the CD4 counts, and transmission risk of individuals that were enrolled in the study were broadly similar to those who were excluded. Similarly, there was no significant difference for the age, sex and ethnicity between the four study periods. However, from 2009 to 2016, there was a significantly higher percentage of MSM compared to 2001–2008. The majority of participants were men (92.9%), and of Han ethnicity (96.2%). Median age was 34 years (interquartile range [IQR] 28–45), and men were younger than women (34 years [IQR 28–45] vs. 37 years [IQR 29–49]). Where available, the overall median baseline CD4 counts was 333 cells per μL (IQR 195–471). The predominant transmission risk groups were MSM (66.7%) or heterosexual contact (26.6%) (Table 1).

**Table 1. Baseline characteristic by sampling phase.**

| | 2001–2008 | 2009–2011 | 2012–2014 | 2015–2016 | Total |
|---|---|---|---|---|---|
| Sex | | | | | |
| Men | 306(85.7) | 461(92.8) | 686(93.8) | 526(96.5) | 1979(92.9) |
| Women | 51(14.3) | 36(7.2) | 45(6.2) | 19(3.5) | 151(7.1) |
| Age at diagnosis(years)[a] | 34(28–42) | 35(28–45) | 34(28–46) | 34(28–47) | 34(28–45) |
| CD4 counts (cells per μL)[b] | 323(200–433) | 329(191–441) | 358(228–519) | 304(162–450) | 333(195–471) |
| Transmission risk group[c] | | | | | |
| Heterosexual | 119(36) | 155(32.0) | 178(24.5) | 115(21.3) | 567(27.2) |
| MSM | 168(50.8) | 304(62.7) | 534(73.5) | 414(76.5) | 1420(68.1) |
| IDU | 23(6.9) | 21(4.3) | 13(1.8) | 12(2.2) | 69(3.3) |
| Blood transfusion | 21(6.3) | 4(0.8) | 2(0.3) | 0(0) | 27(1.3) |
| Mother to child | 0(0) | 1(0.2) | 0(0) | 0(0) | 1(0.05) |
| Subtype | | | | | |
| A1 | 3(0.8) | 4(0.8) | 5(0.7) | 2(0.4) | 14(0.7) |
| B | 151(42.3) | 119(23.9) | 111(15.2) | 69(12.7) | 450(21.1) |
| C | 9(2.5) | 5(1) | 7(1) | 1(0.2) | 22(1) |
| F1 | 1(0.3) | 0(0) | 0(0) | 0(0) | 1(0) |
| G | 3(0.8) | 1(0.2) | 0(0) | 0(0) | 4(0.2) |
| 01_AE | 119(33.3) | 230(46.3) | 378(51.7) | 277(50.8) | 1004(47.1) |
| 02_AG | 0(0) | 0(0) | 3(0.4) | 1(0.2) | 4(0.2) |
| 06_cpx | 3(0.8) | 0(0) | 0(0) | 0(0) | 3(0.1) |
| 07_BC | 56(15.7) | 123(24.7) | 172(23.5) | 140(25.7) | 491(23.1) |
| 08_BC | 7(2.0) | 4(0.8) | 2(0.3) | 3(0.6) | 16(0.8) |
| 55_01B | 0(0) | 3(0.6) | 7(1) | 3(0.6) | 13(0.6) |
| 57_BC | 1(0.3) | 0(0) | 0(0) | 0(0) | 1(0) |
| 59_01B | 0(0) | 0(0) | 2(0.3) | 2(0.4) | 4(0.2) |
| 63_02A1 | 0(0) | 1(0.2) | 0(0) | 0(0) | 1(0) |
| 65_cpx | 1(0.3) | 1(0.2) | 4(0.5) | 6(1.1) | 12(0.6) |
| 67_01B | 0(0) | 1(0.2) | 1(0.1) | 0(0) | 2(0.1) |
| 68_01B | 0(0) | 0(0) | 2(0.3) | 2(0.4) | 4(0.2) |
| URFs | 3(0.8) | 5(1) | 37(5.1) | 39(7.2) | 84(3.9) |
| Ethnicity | | | | | |
| Han | 334(93.6) | 472(95.0) | 705(96.4) | 538(98.7) | 2049(96.2) |
| Minority | 23(6.4) | 25(5.0) | 26(3.6) | 7(1.3) | 81(3.8) |

Data are n (%) or median (IQR).

MSM = Men who have sex with men.

IDU = Injecting drug user

[a]Data for n = 2,125.

[b]Data for n = 1,858.

[c]Data for n = 2,084.

URFs = Unique Recombinant Forms

## Temporal trends of HIV subtypes

The most common HIV subtype and circulating recombinant forms(CRFs) circulating among Beijing residents were CRF01_AE (47.1%), CRF07_BC(23.1%), B(21.1%), and URF(3.9%). Additional clades including subtypes A1, C, F1, CRF02_AG, CRF06_cpx, CRF08_BC, CRF55_01B, CRF57_BC, CRF59_01B, CRF63_02A1, CRF65_cpx, CRF67_01B, and CRF68_01B were present

in less than 1.0% of persons (Fig 1). Table 1 presents the temporal trends for these main subtypes and CRFs. There was a substantial increase in the prevalence of HIV CRF07_BC over time. The prevalence of CRF01_AE increased and stabilized. Interestingly, the prevalence of subtype B continuously declined throughout the period of the study.

## Distribution of subtypes and CRFs

The percentage of subtypes and CRFs circulating in Beijing varied significantly by sex, age, ethnicity, and transmission risk group. Table 2 shows the subtype diversity within

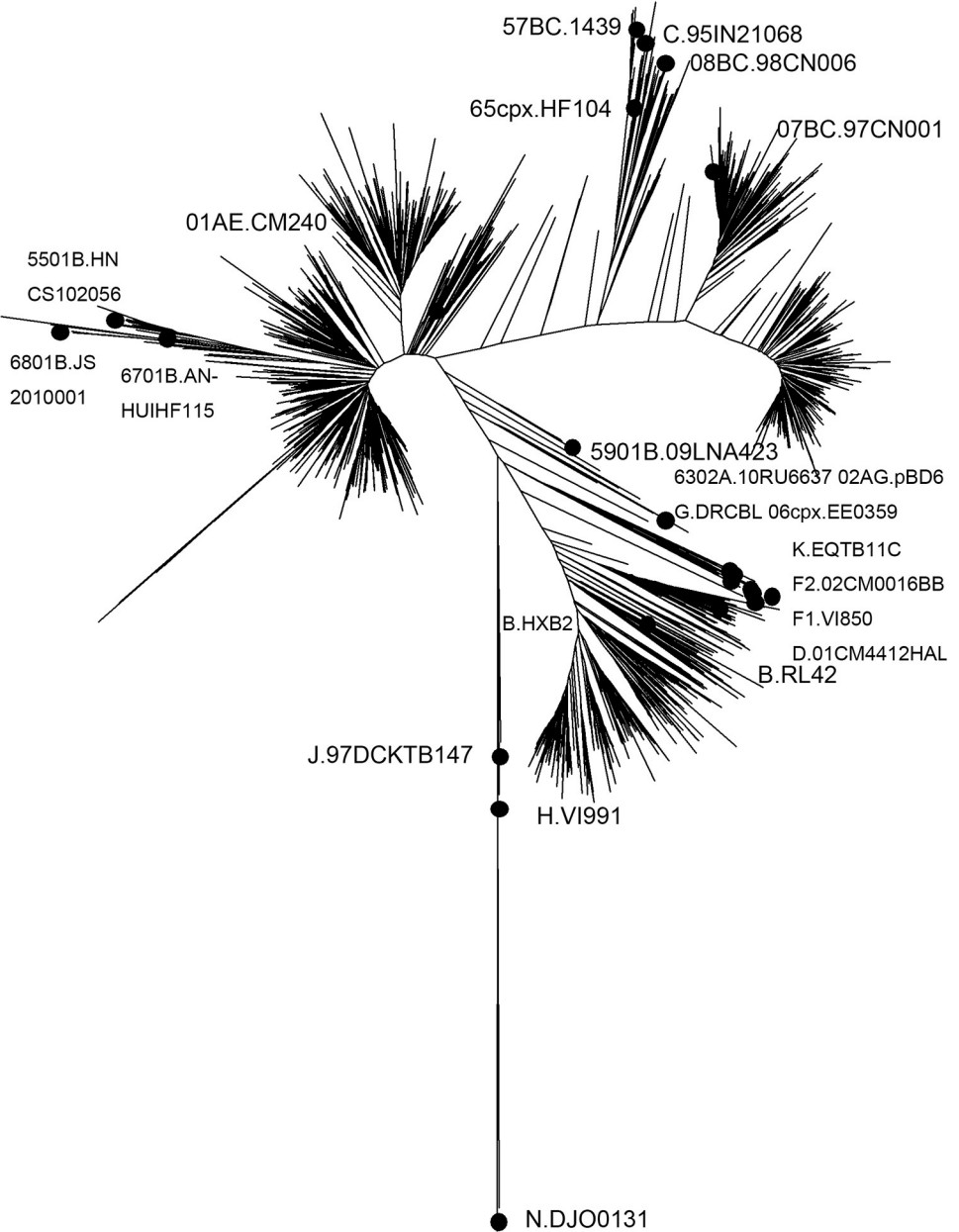

**Fig 1. Phylogenetic analysis of *pol* sequences.** Phylogenetic tree was constructed using neighbor-joining methods (Mega 6.0). The black solid squares indicate reference sequences from the Los Alamos HIV sequence database.

**Table 2. Subtype assignment by selected characteristics.**

|  | CRF01_AE | B | CRF07_BC | URFs | Other |
|---|---|---|---|---|---|
| sex |  |  |  |  |  |
| Men | 959(95.5) | 419(93.1) | 443(90.2) | 78(92.9) | 80(79.2) |
| Women | 45(4.5) | 31(6.9) | 48(9.8) | 6(7.1) | 21(20.8) |
| Age at diagnosis(years) group[a] |  |  |  |  |  |
| <25 | 154(15.4) | 44(9.8) | 67(13.7) | 10(11.9) | 9(8.9) |
| 25–44 | 617(61.5) | 298(66.5) | 259(53) | 48(57.1) | 56(55.4) |
| 45–64 | 198(19.7) | 92(20.5) | 132(27) | 20(23.8) | 31(30.7) |
| 65- | 34(3.4) | 14(3.1) | 31(6.3) | 6(7.1) | 5(5) |
| CD4 counts (cells per μL)[b] |  |  |  |  |  |
| <200 | 252(27.6) | 106(29) | 75(17.6) | 18(23.4) | 22(28.9) |
| 200–349 | 261(28.6) | 101(27.6) | 121(28.3) | 22(28.6) | 14(18.4) |
| 350–499 | 207(22.7) | 86(23.5) | 128(30) | 20(26) | 14(18.4) |
| >499 | 192(21.1) | 73(19.9) | 103(24.1) | 17(22.1) | 26(34.2) |
| Transmission risk group[c] |  |  |  |  |  |
| Heterosexual | 229(23.2) | 134(30.7) | 130(27.0) | 18(22.0) | 56(59.6) |
| MSM | 744(75.2) | 286(65.4) | 300(62.2) | 59(72.0) | 31(33.0) |
| IDU | 13(1.3) | 7(1.6) | 39(8.1) | 5(6.1) | 5(5.3) |
| Blood transfusion | 3(0.3) | 10(2.3) | 12(2.5) | 0(0) | 2(2.1) |
| Mother to child | 0(0) | 0(0) | 1(0.2) | 0(0) | 0(0) |
| Ethnicity |  |  |  |  |  |
| Han | 977(97.3) | 428(95.1) | 471(95.9) | 82(97.6) | 91(90.1) |
| Minority | 27(2.7) | 22(4.9) | 20(4.1) | 2(2.4) | 10(9.9) |

Data are n (%).

[a]Data for n = 2,125.

[b]Data for n = 1,858.

[c]Data for n = 2,084.

URFs = Unique Recombinant Forms.

MSM = Men who have sex with men.

IDU = Injecting drug users.

Other = A1, C, F1, CRF02_AG, CRF06_cpx, CRF08_BC, CRF55_01B, CRF57_BC, CRF59_01B, CRF63_02A1, CRF65_cpx, CRF67_01B, and CRF68_01B.

demographic subgroups. MSM were found to predominantly have CRF01_AE virus. Individuals with heterosexual transmission risk had a much greater variety of HIV, with the most frequent subtype being CRF01_AE. CRF07_BC infections were more common in injecting drug users(IDU). The phylogenetic analysis showed no evidence of laboratory carry-over contamination.

## Prevalence and patterns of TDR

The overall prevalence of TDR was 4.5% (95% CI: 3.6–5.4), with mutations associated with PIs being the most common form of mutation (2.8%, 95% CI:2.1–3.5), followed by mutations associated with NRTIs (1.0%, 95% CI:0.6–1.5) and NNRTIs (0.9%, 95% CI: 0.6–1.4). Dual-class resistance was uncommon (0.28%, 95% CI:0.1–0.6). Women tended to have lower prevalence of TDR than men. In the heterosexual transmission risk group, men had twice the prevalence of TDR than women (4.1%, 95% CI: 2.4–6.1 in men, vs. 1.6%, 95% CI: 0.2–4.6 in women). Notably, the prevalence of TDR in CRF07_BC viruses was significantly lower than in other viral strains. The prevalence of TDR did not differ significantly between transmission risk groups.

The most prevalent mutation associated with PIs TDR, M46L (1.9%, 95% CI: 1.4–2.6), was present in 4.0% (95% CI:2.9–5.3) of the CRF01_AE HIV strains tested. The most frequent mutation associated with NRTIs TDR were the thymidine analogue mutations (TAM), of which the most prevalent were the mutation of M41L (0.2%, 95% CI:0.05–0.4), followed by mutation of M184V(0.28%, 95% CI:0.1–0.6). The K103N, Y181C, and K101E were the most common mutations associated with NNRTIs TDR, which were found in 0.28% (95% CI:0.1–0.6), 0.23% (95% CI:0.07–0.5), and 0.23% (95% CI:0.07–0.5) of individuals, respectively. No polymorphisms and mixture base were found in the TDR position. Of the 95 sequences with at least one TDR mutation, resistance was confined to a single drug class for 89 sequences (93.7%, 95% CI: 87.9–97.7), and 87 sequences (91.6%, 95% CI:85.2–96.3) had a single mutation. Of the 95 sequences, 21(22.1%, 95% CI: 14.4–31.0) had high-level resistance (Stanford 5), 17 (17.9%, 95% CI:11.0–26.2) had intermediate-level resistance (Stanford 4), and 49 (51.6%, 95% CI:41.6–61.6) had low-level resistance (Stanford 3). Seventeen sequences (17.9%, 95% CI: 10.1–26.2) showed loss of susceptibility to NRTIs (AZT 8.4%, 3TC 8.4%, TDF 3.2%), 21 sequences (22.1%, 95% CI:14.4–31.0) to NNRTIs (EFV 22.1%, ETR 13.7%, NVP 22.1%, RPV 14.7%), and 55 sequences (57.9%, 95% CI:47.9–67.6) to PIs (LPV 2.1%, NFV 58.9%).

### Time trends and correlates of TDR

The annual prevalence of TDR in our study ranged from 1.7% to 10.3% of the samples tested. There was no statistically significant decline in the annual trendof TDR over the study period when using the univariable (p = 0.08) or multivariable analysis (p = 0.14) (S1 Fig). There was a significant decline in the prevalence of TDR over between 2001 and 2016 when both the univariable and multivariable analyses were performed (Fig 2, Table 3). Comparing the TDR by the ARTclasses, the PIs followed the same time trend as the overall prevalence (p = 0.0003),

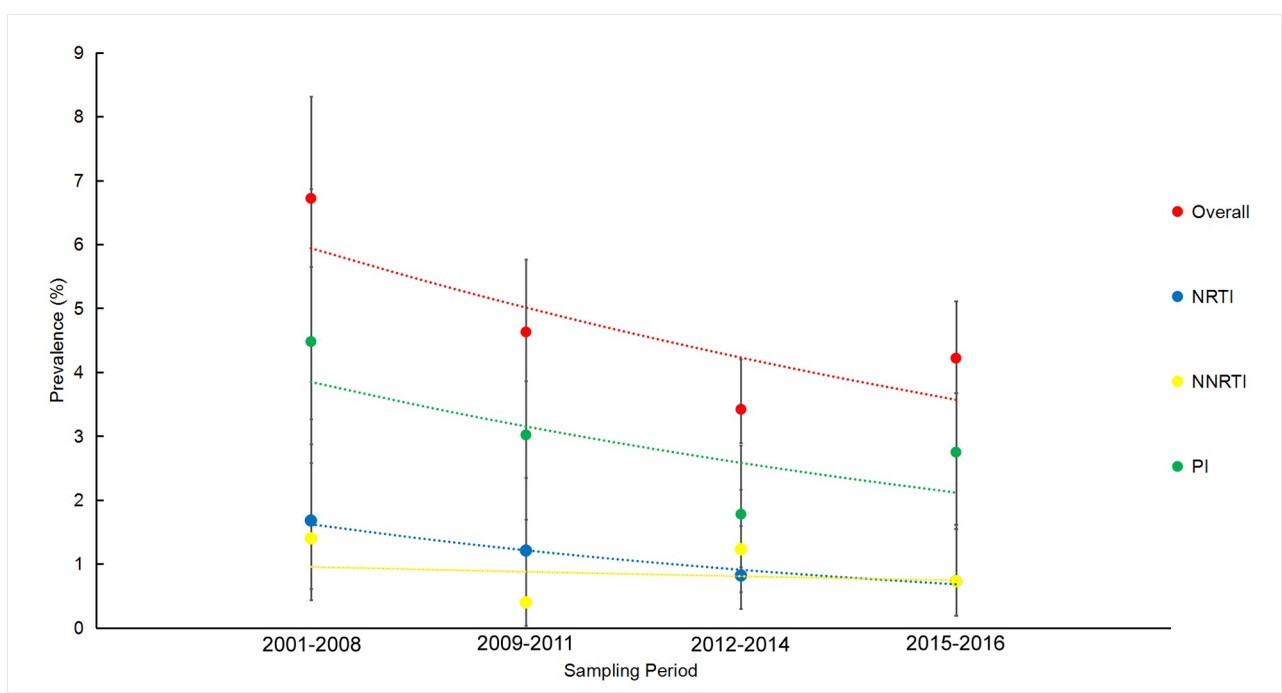

**Fig 2. Temporal trends of prevalence of transmitted drug resistance by sampling phase.** Vertical bars = 95% CI; The trend line is predicted prevalence of transmitted drug resistance; NRTIs = nucleoside reverse transcriptase inhibitors; NNRTIs = non-NRTIs; PIs = protease inhibitors.

**Table 3. Demographic and clinical factors associated with TDR.**

| | Number of sequences | Prevalence of TDR[#] | Univariable analysis[*] | | Multivariable analysis† | |
|---|---|---|---|---|---|---|
| | | | OR (95% CI) | p value | OR (95% CI) | p value |
| Sex | | | | | | |
| Men | 1979 | 93(4.7) | Reference | | | |
| Women | 151 | 2(1.3) | 0.27(0.04–0.87) | 0.07 | 0.26(0.04–0.85) | 0.06 |
| Age at diagnosis(years) group[a] | | | | | | |
| <25 | 284 | 13(4.6) | Reference | | | |
| 25–44 | 1278 | 59(4.6) | 1.01(0.56–1.95) | 0.98 | | |
| 45–64 | 473 | 17(3.6) | 0.78(0.37–1.66) | 0.5 | | |
| 65- | 90 | 4(4.4) | 0.97(0.27–2.82) | 0.96 | | |
| CD4 counts (cells per μL)[b] | | | | | | |
| <200 | 473 | 20(4.2) | Reference | | | |
| 200–349 | 519 | 23(4.4) | 1.05(0.57–1.95) | 0.88 | | |
| 350–499 | 455 | 18(4.0) | 0.93(0.48–1.79) | 0.83 | | |
| >499 | 411 | 22(5.4) | 1.28(0.69–2.40) | 0.43 | | |
| Transmission risk group[c] | | | | | | |
| Heterosexual | 567 | 20(3.5) | Reference | | | |
| MSM | 1420 | 67(4.7) | 1.35(0.83–2.31) | 0.24 | | |
| IDU | 69 | 4(5.8) | 1.68(0.48–4.61) | 0.36 | | |
| Blood transfusion | 27 | 1(3.7) | 1.05(0.06–5.36) | 0.96 | | |
| Ethnicity | | | | | | |
| Han | 2049 | 94(4.6) | Reference | | | |
| Minority | 81 | 1(1.2) | 0.26(0.01–1.19) | 0.18 | | |
| Sampling Phase | | | | | | |
| 2001–2008 | 357 | 24(6.7) | Reference | | | |
| 2009–2011 | 497 | 23(4.6) | 0.58(0.33–1.01) | 0.05 | 0.63(0.34–1.16) | 0.13 |
| 2012–2014 | 731 | 25(3.4) | 0.59(0.35–1.02) | 0.05 | 0.43(0.23–0.79) | 0.006 |
| 2015–2016 | 545 | 23(4.2) | 0.51(0.19–1.2) | 0.15 | 0.54(0.29–0.999) | 0.048 |
| Subtype | | | | | | |
| 01_AE | 1004 | 61(6.1) | Reference | | | |
| 07_BC | 491 | 5(1) | 0.16(0.06–0.36) | <0.01 | 0.16(0.06–0.37) | 0.0001 |
| B | 450 | 19(4.2) | 0.68(0.39–1.13) | 0.15 | 0.58(0.33–0.98) | 0.05 |
| URFs | 84 | 4(4.8) | 0.77(0.23–1.94) | 0.63 | 0.88(0.26–2.24) | 0.81 |
| Others | 101 | 6(5.9) | 0.98(0.37–2.15) | 0.96 | 0.99(0.37–2.22) | 0.99 |

MSM = Men who have sex with men.

IDU = Injecting drug user

OR = odds ratio.

[*]Univariable logistic regression analysis.

†Multivariable logistic regression analysis.

[#]Data are n (%).

[a]Data for n = 2,125.

[b]Data for n = 1,858.

[c]Data for n = 2,084.

TDR = transmitted drug resistance.

Other = A1, C, F1, CRF02_AG, CRF06_cpx, CRF08_BC, CRF55_01B, CRF57_BC, CRF59_01B, CRF63_02A1, CRF65_cpx, CRF67_01B, and CRF68_01B.

URFs = Unique Recombinant Forms.

but there was no significant change in time trend for NRTIs or NNRTIs (p = 0.34 for NRTIs, p = 0.37 for NNRTIs) (Fig 2).

Multivariable analysis revealed association between TDR, and HIV subtype and sampling phase, with risk reduced for CRF07_BC and phase 2012–2016, compared to CRF01_AE and phase 2001–2008 (Table 3). In two sensitivity analyses, which included individuals younger than 18 years old, and excluded individual transmission risk group respectively, the magnitude of the associations did not change significantly(data not shown).

### CD4 counts data missing

Because the rate of missing CD4 counts data was relatively high (12.8%), multiple imputation was used in the logistics analysis(S2 Table). In addition, four sensitivity experiments were carried out by excluding individual sampling phase (S3–S6 Tables). Indeed, neither the multiple imputation method nor the sensitivity experiments proved that CD4 counts was associated with TDR.

## Discussion

This study prospectively analyzed nucleotide and amino acid sequences to decipher the temporal trends in prevalence of TDR and the genetic diversity of HIV among 2,130 Beijing residents. A high degree of viral diversity was observed with multiple subtypes and CRFs among Beijing residents. CRF01_AE, CRF07_BC, and subtype B were the most common clades circulating among Beijing residents. The trends for CRF01_AE and CRF07_BC increased over time, whereas B had a decreasing trend. A similar trend was observed in other provinces across China [15–17]. Indeed, Beijing is a popular destination for floating populations that come from other provinces and other countries. It is likely that the high HIV genetic diversity observed in this study population could be due to the influx of non-residents and viral lineages that circulate in other provinces or other countries [18].

The overall prevalence of TDR among residents newly diagnosed with HIV infection in Beijing was low. There was a apparent declining trend during study period, which was consistent with the results of other molecular diversity studies in other provinces of China [15–17,19,20]. There was no significant difference in the prevalence of TDR when comparing the sex, the age, the transmission risk groups and the ethnicity of the study population. Of the three main clades (CRF01_AE, CRF07_BC, and subtype B), CRF07_BC had the lowest prevalence of TDR. This prevalence was significantly lower than reported in Mexico, San Diego (USA), and Europe[21–23]. The low prevalence of TDR is most likely due to a short period exposure to antiretroviral drugs. It is only in 2003 that the implementation of NFATP was widely applied in Beijing. Indeed, Beijing has a relatively shorter experience using ART compared to North American and Europe, which started using ART in middle of 1990. Results from this study also indicated there was high prevalence of TDR for PIs and a low prevalence of TDR for NRTIs and NNRTIs. This was unexpected, given that NRTIs and NNRTIs are widely used in Beijing as first-line treatment and the Lopinavir/Ritonavir is the only ART drug prescribed as a second-line regimen. The higher prevalence of PIs can be attributed to an unexpectedly high proportion of participants with CRF01_AE virus that harbored M46L mutation, which could cause low-level drug resistance to nelfinavir (NFV). Because NFV is not prescribed in China, this high prevalence has little practical meaning.

Beijing is an international cosmopolitan city and is a human mobility hub which maintains a very intense movement of people, from both within China and overseas. People move to Beijing because they are attracted to employment, medical need, and tourism. Three quarter of the 21,886 individuals with HIV infection in Beijing are floating population [1]. Often, it is

commonly accepted that the floating population dominated the Beijing HIV epidemic. However, could IDUs infected with HIV in Sinkiang or former blood donors in Henan province, for instance, truly represented the epidemic of Beijing? Being diagnosed in Beijing does not necessarily mean that the infection occurred in Beijing. If patients were infected in their home province, though diagnosed in Beijing, they actually reflected the epidemic of their hometown. Moreover, floating population and residents are different groups of people, with the former coming from across China and the latter only from Beijing. Floating populations come and go in Beijing, but residents will always be there. Therefore we suggested that floating population most likely represent imported HIV epidemic, while residents represent ongoing transmission of HIV and could better represent the epidemic in Beijing.

The data presented in this study sheds light and provides new insights to better understand the molecular epidemiology of HIV and will assist in the development of the prevention and the treatment strategies for the control of the HIV/AIDS epidemic in Beijing and beyond. Because most patients generally respond well to first-line regimens, the routine genotype testing is not required prior to treatment. There are no pressing needs for expensive second-line regimens. Although NFV is not used as part of the first-line ART regimens in Beijing, it is worth noting that there are at least a small number (1.9%) of people harbor viruses with TDR to these drugs. Thus, the use of NFV in Beijing should be examined with caution. Vaccine designer in Beijing should take the fact that CRF01_AE, CRF07_BC and subtype B constituted more than 90% of all the clades into consideration in select appropriate candidate HIV strains. Viral load (VL) kit manufacturer should also know this in designing the primer to ensure accurate HIV RNA quantitation of non-B subtypes in Beijing. Interestingly, as shown in Table 3, the odds ratio and p value for women was inconsistent. We speculated that the sample size of women is too small to discern the significant difference in this study. Our recent national survey with larger sample size of women confirms that our speculation is correct [24].

To our knowledge this is the largest study to cover the longest-period (16 years) on HIV subtypes and TDR in Beijing. We analyzed sequences representing half of all known residents cases of HIV infection that were diagnosed in Beijing during 2001–2016, which allowed us to carry out this analysis with reasonable accuracy.

However, several limitations are worth mentioning. Firstly, the study population was limited for women. Secondly, VL information was not available, which did not permit the evaluation of the association between VL and TDR.

In summary, this study of 2,130 HIV infected patients show that there is a high genetic heterogeneity of HIV in Beijing than previously appreciated. However, the prevalence of TDR was low with a declining trend over nearly a two decades period. The prevalence of TDR was lower in individuals infected with CRF07_BC than those infected with CRF01_AE. The widespread distribution of ART did not necessarily lead to an increase of TDR. To better formulate a more efficacious response policy on HIV/AIDS in a heterogenic city such as Beijing, residents and floating population should be analyzed separately.

## Supporting information

**S1 Fig. Temporal trends of overall prevalence of transmitted drug resistance by sampling year.** Vertical bars = 95% CI. The trend line is predicted overall prevalence of transmitted drug resistance.
(TIF)

**S1 Table. Demographic characteristics of individuals with genotype vs. those with non-genotype.**
(DOCX)

**S2 Table. Univariable logistic regression analysis of CD4 counts associated with transmitted drug resistance with multiple imputation.**
(DOCX)

**S3 Table. Sensitivity analysis of CD4 counts associated with transmitted drug resistance by excluding sampling phase 2001–2008.**
(DOCX)

**S4 Table. Sensitivity analysis of CD4 counts associated with transmitted drug resistance by excluding sampling phase 2009–2011.**
(DOCX)

**S5 Table. Sensitivity analysis of CD4 counts associated with transmitted drug resistance by excluding sampling phase 2012–2014.**
(DOCX)

**S6 Table. Sensitivity analysis of CD4 counts associated with transmitted drug resistance by excluding sampling phase 2015–2016.**
(DOCX)

## Acknowledgments

We thank the study participants and the staff at the collaborating clinical sites and laboratories. We thank the local health workers of the BHLN, who spent numerous hours and great effort in obtaining, verifying, and cleaning the data used in this study. We are grateful to Dr. John Mokili, from Viral Information Institute at San Diego State University for sharing his expertise on Molecular Epidemiology of HIV/AIDS and for critical review and insightful editing of the manuscript.

## Author Contributions

**Conceptualization:** Jingrong Ye, Hui Xing, Yi Feng, Ruolei Xin, Yuhua Ruan, Yiming Shao, Hongyan Lu.

**Data curation:** Jingrong Ye, Mingqiang Hao, Juan Wang, Ji Zeng, Jing Chen.

**Formal analysis:** Jingrong Ye, Mingqiang Hao.

**Funding acquisition:** Hui Xing, Yi Feng, Ruolei Xin, Yuhua Ruan, Yiming Shao, Hongyan Lu.

**Investigation:** Juan Wang, Ji Zeng, Jing Chen, Hongyan Lu.

**Methodology:** Jingrong Ye, Mingqiang Hao, Yuncong Wang, Shuai Zhao, Yinxiao Hao, Xue Li.

**Project administration:** Hui Xing, Yiming Shao.

**Resources:** Hui Xing, Ruolei Xin, Yuhua Ruan, Hongyan Lu.

**Software:** Jingrong Ye, Mingqiang Hao.

**Supervision:** Jingrong Ye, Hui Xing, Yi Feng, Ruolei Xin, Yiming Shao, Hongyan Lu.

**Validation:** Hui Xing.

**Visualization:** Jingrong Ye, Mingqiang Hao.

**Writing – original draft:** Jingrong Ye.

**Writing – review & editing:** Jingrong Ye, Mingqiang Hao, Yi Feng, Yiming Shao, Hongyan Lu.

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
