## [Decision Letter · Decision Letter 0]

9 Sep 2019

PONE-D-19-15622

Characterization of HIV diversity and drug resistance in Beijing, China, 2001-2016

PLOS ONE

Dear Dr. Lu,

Thank you for submitting your manuscript to PLOS ONE. After careful consideration, we feel that it has merit but does not fully meet PLOS ONE’s publication criteria as it currently stands. Therefore, we invite you to submit a revised version of the manuscript that addresses the points raised during the review process.

We would appreciate receiving your revised manuscript by Oct 24 2019 11:59PM. To enhance the reproducibility of your results, we recommend that if applicable you deposit your laboratory protocols in protocols.io, where a protocol can be assigned its own identifier (DOI) such that it can be cited independently in the future. For instructions see: http://journals.plos.org/plosone/s/submission-guidelines#loc-laboratory-protocols

We look forward to receiving your revised manuscript.

Kind regards,

Jason Blackard, PhD

Academic Editor

PLOS ONE

Journal Requirements:

2. Our internal editors have looked over your manuscript and determined that it is within the scope of our Antimicrobial Resistance call for papers. This collection of papers is headed by a team of Guest Editors for PLOS ONE: Kathryn Holt (Monash University and London School of Hygiene and Tropical Medicine), Alison H. Holmes (Imperial College London), Alessandro Cassini (WHO Infection Prevention and Control Global Unit), Jaap A. Wagenaar (Utrecht University). The Collection will encompass a diverse range of research articles; additional information can be found on our announcement page: https://collections.plos.org/s/antimicrobial-resistance. If you would like your manuscript to be considered for this collection, please let us know in your cover letter and we will ensure that your paper is treated as if you were responding to this call. If you would prefer to remove your manuscript from collection consideration, please specify this in the cover letter.

Additional Editor Comments (if provided):

This a longitudinal study of HIV subtypes and drug resistance in China.

The methods and analyses are sound.  However, the manuscript requires careful revision by a native English speaker or a professional editing service.

The population size is quite large although heavily skewed towards men.  Does this reflect the gender distribution of HIV in China?

Lines 69 and 80:  what is “floating population”?

Reviewers' comments:

Reviewer's Responses to Questions

**Comments to the Author**

1. Is the manuscript technically sound, and do the data support the conclusions?

Reviewer #1: Partly

Reviewer #2: Partly

2. Has the statistical analysis been performed appropriately and rigorously? 

Reviewer #1: Yes

Reviewer #2: No

3. Have the authors made all data underlying the findings in their manuscript fully available?

Reviewer #1: Yes

Reviewer #2: No

4. Is the manuscript presented in an intelligible fashion and written in standard English?

Reviewer #1: No

Reviewer #2: No

5. Review Comments to the Author

Reviewer #1: The investigators have analysed large size of drug resistance data and adds value to the medical literature substantially. Few things to be considered before publishing this manuscript:

1) The manuscript English could be improved as much as possible.

2) Atleast a brief mention of laboratory methodology and instrumentation is needed. Eg. Instruments used for CD4 count, Gene sequencing, etc.,

3) The study conducted with the samples collected between 2001 - 2016, but the break up used could be pattern based rather than convenient sake. Eg. 2001-2004, 2005-2008, 2009-2012, 2012-2015 or so.

4) The drug resistance patterns also could be analysed with the year break up with the pattern to observe the trends better.

5) Polymorphisms and mixtures in the drug resistance positions have not been addressed. If not observed, that could be mentioned.

6) Looks like the study did not include plasma viral load (PVL) data and if so, that could be listed in the study limitation as this could potentially affect the study data. If PVL is available, IQR for the available data could be mentioned.

7) Figure 2 is not readable.

Reviewer #2: The manuscript by Lu and colleagues describes high HIV genetic diversity and declining trends in the prevalence of overall and some class-specific HIV drug resistance in Beijing between 2001-2016. While the high genetic diversity of HIV is well known globally, the observed declines in HIV drug resistance in Beijing are of relevance and contrast trends observed in other settings. The discussion of the data to potentially explain these findings is well-balanced. However, some aspects of the methods and presentation of results would benefit from further clarification. Despite being a national data source, increased transparency on potential selection bias is strongly recommended. The manuscript would also benefit from additional grammatical review. Below are specific comments:

Title:

The title of the manuscript should included further specification of the study population (i.e., newly diagnosed residents).

Abstract:

Line 38 – The outcome for the statement, “The overall prevalence was 4.5%”, should be specified.

Introduction:

Line 69 – It is unclear at first mention what a floating population is.

Methods:

Line 96 – The manuscript states that all data were collected and analyzed in the course of routine public health surveillance, but does this include retrospective analysis of stored specimen? Were patients notified if they had drug resistant strains?

Line 104 – It needs to be specified that it is 21,886 individuals ever diagnosed with HIV infection

Line 110-11: It is unclear what “standardized sampling strategy” means and why only half of the samples from all newly identified individuals were included in the study. Further clarification is needed.

Line 153 – The term “multivariable” would be more appropriate than “multivariate”, consistent with what is used later in the manuscript (e.g., Line 159).

Line 162 – Missing data were handled by list-wise deletion but how much missing data were there and for which variables? Was missing data on covariates associated with the outcome or covariates examined? The final sample size of the multivariable analyses should be included as a footnote in corresponding tables.

Results:

Lines 170-176 – Although a statement is provided comparing characteristics of individuals who were included in and excluded from the analytic sample, these data should be reported, perhaps as supplementary tables.

Table 2 is difficult to interpret and the total column repeats data that was presented in Table 1. It should be noted whether column or row percentages are presented and only one type should be provided in a given table. Table 2 currently seems to use both.

Lines 266-267 – The line “lower drug prevalence” is too vague.

Lines 267-268 – The line “men had twice the prevalence of women” should specify the outcome. Prevalence of what?

Line 290 – It is incorrect to state “this trend became significance when controlling for potential confounders”, as the trend was already statistically significant in univariate analysis.

Line 331 (Table 3) – Why does the footnote for Table 3 specify “multilevel” regression was used? This contrasts what was written in the methods section.

Line 333 – Related to the missing data issue discussed above, there were at least 501 individuals (24% of 2,130) who were dropped from the multivariable analysis due to missing data on CD4 count. More rigorous methods to handle missing data should be explored or an explanation for why they were not pursued should be explained.

Discussion

Line 337 – It is unclear how this is a “prospective” study if sequences prior to routine genotyping were examined for TDR retrospectively.

Lines 389-392 – The discussion states that the data presented in this study “[assist] in the development of prevention and treatment strategies to influence the HIV epidemic within Beijing.” Further elaboration on how would be helpful.

Nonresponse should be discussed in further detail.

6. PLOS authors have the option to publish the peer review history of their article (what does this mean?). If published, this will include your full peer review and any attached files.

Reviewer #1: No

Reviewer #2: No

---

## [Author Response · Author response to Decision Letter 0]

18 Nov 2019

Response to reviewers

Dear Editor and Reviewers:

We would like to thank Plos One for giving use the opportunity to revise our manuscript. We thank reviewers for their careful read and thoughtful comments on previous draft. We have carefully taken their comments into consideration in preparing our revision. Below is our response to their comments.

Journal Requirements:

Yes, we are sure that our manuscript meets Plos One’ style requirements.

2. Our internal editors have looked over your manuscript and determined that it is within the scope of our Antimicrobial Resistance call for papers. This collection of papers is headed by a team of Guest Editors for PLOS ONE: Kathryn Holt (Monash University and London School of Hygiene and Tropical Medicine), Alison H. Holmes (Imperial College London), Alessandro Cassini (WHO Infection Prevention and Control Global Unit), Jaap A. Wagenaar (Utrecht University). The Collection will encompass a diverse range of research articles; additional information can be found on our announcement page: https://collections.plos.org/s/antimicrobial-resistance. If you would like your manuscript to be considered for this collection, please let us know in your cover letter and we will ensure that your paper is treated as if you were responding to this call. If you would prefer to remove your manuscript from collection consideration, please specify this in the cover letter.

Yes, we are greatly pleased that our manuscripts could be considered for this collection.

Yes, we included the full name of the research ethics committee that approved our study. We added the same text to the “Ethnics Statement” field of the submission form.

Additional Editor Comments (if provided):

This a longitudinal study of HIV subtypes and drug resistance in China.

 The methods and analyses are sound. However, the manuscript requires careful revision by a native English speaker or a professional editing service.

Yes, we invited Dr. John Mokili, from Viral Information Institute at San Diego State University for critical review and insightful editing of the manuscript.

 The population size is quite large although heavily skewed towards men. Does this reflect the gender distribution of HIV in China?

Yes, this is exactly real gender distribution of HIV in Beijing, the capital of China. Of the 17,421 individuals living with HIV in Beijing by the end of 2017, 95.1% were men. The gender ratio of individuals with HIV varied greatly between provinces in China. The national gender ratio of men vs. women of people with HIV is 3.7:1 in 2016.

Lines 69 and 80: what is “floating population”?

To get a full appreciation of what “floating population” means we must turn first to Hukou system, the basic system of household registration in China. The Hukou system officially identifies a person as a resident of an area and includes identifying information such as name, parents, spouse, and date of birth. The Hukou system is of critical importance to people in China. Someone without Hukou is regarded as an illegal resident. The Hukou is analogous to the US Permanent Resident Card, but it has a more important meaning to Chinese people. Hukou is the system of household registration used in China. The system itself is more properly called "Huji", and has origins in ancient China. Due to its connection to social programs, which assigns benefits based on agricultural and non-agricultural residency status (often referred to as rural and urban), the Hukou system is sometimes likened to a form of caste system.

Floating population is a terminology used to describe a group of people who reside in a given population for a certain amount of time and for various reasons, but are not generally considered part of the official census count [1].

In our study, we categorized the residents and floating population based on Hukou system. We defined patients with Beijing Hukou as residents, and those without Beijing Hukou as floating population. To make two extreme instances, if Chinese people with Beijing Hukou lived in U.S for 20 years or more, we still regard them as Beijing residents, unless they checked out the Beijing Hukou. In the contrary class, if people without Beijing Hukou (certainly have the local Hukou, every Chinese people have Hukou) lived in Beijing for 20 years or more, we still call them floating population. Of course, people in these two instances were minority. Because Hukou is always associated with ware fare, education, medical services, old age housekeeping, most Chinese people live their most of their life in the place where their Hukou belong. Moreover, in China, no one could always float, they will reside in the end.

1.http://wikipedia.moesalih.com/Floating_population.

Reviewers' comments:

Reviewer's Responses to Questions

Comments to the Author

1. Is the manuscript technically sound, and do the data support the conclusions?

Yes, our manuscript concerned two of the most important topics of HIV in Beijing: the subtype and the TDR, both of which have important implications. We believe our finding has the potential to transform current understanding of the relationship between antiretroviral therapy (ART) rollout and prevalence of transmitted drug resistance(TDR). There has long been a concern that TDR has increased in parallel with the increasing availability of antiretroviral treatment (ART), which can compromise the effectiveness of first-line ART regimens. The most significant finding of our study is that we observed a declining trend of TDR. This is amazing. Why? Because the reasons for the declining trends in prevalence of TDR in Beijing would also apply to other regions. Our findings provide hope to individuals with HIV on ART in regions with high TDR burden, and serve as an indicator of the likely future trend.

Reviewer #1: Partly

Reviewer #2: Partly

2. Has the statistical analysis been performed appropriately and rigorously? 

Reviewer #1: Yes

Reviewer #2: No

Yes, the statistical analysis was performed using R based on the book by Robert I Kabacoff, “R in Action: Data Analysis and Graphics with R”. Patient age was discretized (18-24, 25-44,154 45-64, and ≥65 years) and CD4 cell count was discretized (<200, 200-349, 155 350-499,and ≥500 cells per μL). Variables were analyzed independently and only those that were associated (p<0.1) with the outcome in the multivariable model were included in the final analysis. Multivariable logistic regression was used to explore the relation TDR and sex, age, ethnicity, HIV subtype, CD4 cell count, transmission risk group, and sampling phase. The analysis to estimate the trend of TDR over time was examined. Using a more rigorous method- the Multiple Imputation(MI)- missing data were analyzed to reduce bias. What was possible has been done. In short, we believe that we have winkled out all the useful meanings beneath these materials.

 3. Have the authors made all data underlying the findings in their manuscript fully available?

Yes, we deposited all our sequences in the HIV databases. We are very happy to share our study with our colleagues not only in China, but also from the world. We are willing to share our finding in detail in all kind of formulation.

Reviewer #1: Yes

Reviewer #2: No

 4. Is the manuscript presented in an intelligible fashion and written in a standard English?

Yes, we invited Dr. John Mokili, from Viral Information Institute at San Diego State University for critical review and insightful editing of the manuscript.

Reviewer #1: No

Reviewer #2: No

5. Review Comments to the Author

Reviewer #1: The investigators have analysed large size of drug resistance data and adds value to the medical literature substantially. Few things to be considered before publishing this manuscript:

1) The manuscript English could be improved as much as possible.

Yes, as any non-native English speakers, we invariably encounter great difficult in writing manuscript in standard English. However, this will not pose any problem. Because we invited Dr. John Mokili, from Viral Information Institute at San Diego State University for critical review and insightful editing of the manuscript.

2) At least a brief mention of laboratory methodology and instrumentation is needed. Eg. Instruments used for CD4 count, Gene sequencing, etc.,

Yes, we described the instruments in Methods section.

3) The study conducted with the samples collected between 2001 - 2016, but the break up used could be pattern based rather than convenient sake. Eg. 2001-2004, 2005-2008, 2009-2012, 2012-2015 or so.

Yes, we established four sampling phases for the convenient sake: 2001-2008, 2009-2011, 2012-2014, and 2015-2016.

4) The drug resistance patterns also could be analysed with the year break up with the pattern to observe the trends better.

Yes, we analyzed the TDR patterns with the year break-up, however, the declining trend lost the significance. Small sample size may be the reason. So we balanced the visuality and statistic significance and established four sampling phases. We also provided the annual prevalence of TDR in supplementary results. Even in annual analysis, the sample before 2005 was too small, so we group them together.

5) Polymorphisms and mixtures in the drug resistance positions have not been addressed. If not observed, that could be mentioned.

Yes, we did not identify polymorphisms and mixture base in the TDR position. We mentioned that in our manuscript.

6) Looks like the study did not include plasma viral load (PVL) data and if so, that could be listed in the study limitation as this could potentially affect the study data. If PVL is available, IQR for the available data could be mentioned.

Yes, we did not include the plasma viral load (PVL) data. We listed this in the study limitation. Indeed, we also maintained a treatment database, which include baseline VL and follow up VL record. In the future study, we will link the baseline VL using the unique ID card number. Now the condition is still not ripe.

7) Figure 2 is not readable.

Yes, we rewrote the Figure 2.

Reviewer #2: The manuscript by Lu and colleagues describes high HIV genetic diversity and declining trends in the prevalence of overall and some class-specific HIV drug resistance in Beijing between 2001-2016. While the high genetic diversity of HIV is well known globally, the observed declines in HIV drug resistance in Beijing are of relevance and contrast trends observed in other settings. The discussion of the data to potentially explain these findings is well-balanced. However, some aspects of the methods and presentation of results would benefit from further clarification. Despite being a national data source, increased transparency on potential selection bias is strongly recommended. The manuscript would also benefit from additional grammatical review. Below are specific comments:

Title:

The title of the manuscript should included further specification of the study population (i.e., newly diagnosed residents).

Yes, we rewrote the title.

Abstract:

Line 38 – The outcome for the statement, “The overall prevalence was 4.5%”, should be specified.

Yes, we specified the overall prevalence of TDR by ART class, ie. NRTIs, NNRTIs, and PIs.

Introduction:

Line 69 – It is unclear at first mention what a floating population is.

Yes, we explained the floating population briefly at first mention and explained it in detail in method.

Methods:

Line 96 – The manuscript states that all data were collected and analyzed in the course of routine public health surveillance, but does this include retrospective analysis of stored specimen? Were patients notified if they had drug resistant strains?

Yes, our study included retrospective analysis of the stored specimen. The study period was divided into three phase based on the history of TDR test in Beijing. Before 2006, when an in-house TDR test was first introduced to Beijing the analysis of the samples collected before that was performed retrospective. The patients were not notified of the TDR result. Between 2007-2012, TDR test was mainly for public health purposes. The results of the TDR test were occasionally sent to the doctors in charge of diagnosis of the patients. We are not very sure whether the patients knew their result. Until then, the guideline for testing for TDR in conjunction with ART treatment needed to be reviewed. During 2001-2007, HIV patients with CD4 counts at 200 cells per μL or less were eligible for free ART. In 2008, the CD4 count threshold for treatment was increased to 350 cells per μL. In this context, the results of the TDR were given to all including the newly diagnosed patients if they met the criterion for treatment. Thirdly, from 2013 to date, we provide free routine TDR test for about half of newly diagnosed patients. All patients are informed of their test results. In the future, we plan to increase the coverage to provide free TDR test to all the patients.

Line 104 – It needs to be specified that it is 21,886 individuals ever diagnosed with HIV infection

Yes, we specified the 21,886 individuals ever diagnosed, not the patients living with HIV.

Line 110-11: It is unclear what “standardized sampling strategy” means and why only half of the samples from all newly identified individuals were included in the study. Further clarification is needed.

Yes, our standardized sampling strategy is a simply sampling strategy. Our concept is “the simpler, the better”. In the new revision, we described it as the simple sampling strategy. For consistency, we simply collected about half of the newly identified patients. We believed the 50% sampling rate is high enough to represent the HIV epidemiology in Beijing. We have to consider the cost-effectiveness of the surveillance program. Each TDR test cost us about 50 U.S dollars. So this article worth 100,000 US dollars, the sum of the money of a Mercedes-Benz. This was so dear for us. In the future, we plan to extend the TDR test to all the newly diagnosed patients.

Line 153 – The term “multivariable” would be more appropriate than “multivariate”, consistent with what is used later in the manuscript (e.g., Line 159).

Yes, we used the “multivariable”.

Line 162 – Missing data were handled by list-wise deletion but how much missing data were there and for which variables? Was missing data on covariates associated with the outcome or covariates examined? The final sample size of the multivariable analyses should be included as a footnote in corresponding tables.

In this revision, we used Multiple Imputation(MI) to handle miss data. The missing data (CD4) was not significantly associate with the outcome in univariable analysis (p=0.0969), but it met the criterion for entering multivariable analysis (p<0.1). We provided the final sample size of the multivariable analyses as a footnote in table 3.

Results:

Lines 170-176 – Although a statement is provided comparing characteristics of individuals who were included in and excluded from the analytic sample, these data should be reported, perhaps as supplementary tables.

Yes, we provided the comparison as supplementary S1 Table. 

Table 2 is difficult to interpret and the total column repeats data that was presented in Table 1. 

It should be noted whether column or row percentages are presented and only one type should be provided in a given table. Table 2 currently seems to use both.

Yes, we realize this. We should read the manuscript several more times before submission. We shouldn’t make such a stupid mistake. Please accept my sincere apology for my negligence in writing the Table 1 and 2. We omitted the total column of Table 2, because it repeated data presented in table 1.

Lines 266-267 – The line “lower drug prevalence” is too vague.

Yes, we describe the lower prevalence in details.

Lines 267-268 – The line “men had twice the prevalence of women” should specify the outcome. Prevalence of what?

Yes, it should be “Prevalence of the overall prevalence of TDR”.

Line 290 – It is incorrect to state “this trend became significance when controlling for potential confounders”, as the trend was already statistically significant in univariate analysis.

Sorry, this is my negligence. We described this as “There was a significant decline in the prevalence of TDR over between 2001 and 2016 when both the univariable and multivariable analyses were performed”.

Line 331 (Table 3) – Why does the footnote for Table 3 specify “multilevel” regression was used? This contrasts what was written in the methods section.

Please forgive me for my negligence. We corrected this. The “multilevel” should be “mutivariable”.

Line 333 – Related to the missing data issue discussed above, there were at least 501 individuals (24% of 2,130) who were dropped from the multivariable analysis due to missing data on CD4 count. More rigorous methods to handle missing data should be explored or an explanation for why they were not pursued should be explained.

Thank you very much. Of all the comments, this is the most difficult and the most challenging one. The CD4 count was not significantly associate with TDR in univariable analysis (p=0.0969), but it met the criterion for entering multivariable analysis (p<0.1). To reply this comment, we did three things. 

Firstly, we analyzed the missing of CD4 count by sampling periods and found that CD4 data missing mainly occurred in patients that were diagnosed during 2001-2008 and 2015-2016(Table 1).In early stage, the CD4 missing was easy to understand. At that time, the HIV epidemiology database was just established and in its commencement. There was possibility that some CD4 counts record were not input into the database. Moreover, we still did not begin to provide the free CD4 test. Many patients might not take the CD4 test. The CD4 miss during 2015-2016 should mainly attribute to “HIV/AIDS information leak Affaire in 2016”.In July, 2016, HIV/AIDS individuals’ private information in China was leaked, a lot of illegal events related to HIV/AIDS individuals occurred, for instance, the telecommunication fraud. Hence, the government of China has required the strict management of HIV/AIDS individuals’ information according to Regulations on AIDS prevention and control and Law of Infectious disease prevention and control. The maintaining of national HIV epidemiological database was discontinuedfor nearly a year. During that time, the CD4 count data were not recorded in the database. Now, though the database has resume normal, the missing CD4 has not yet been added into the database. We are sure that most of the individuals diagnosed during 2015-2016 have CD4 counts records. These records are stored in the HIV laboratory. 

Secondly, we do our best to trace these CD4 records and re-input it into the database. Our colleagues Juan Wang and Jing Chen took three weeks to finish the job-501 records, one by one. After these efforts, we get back 229 CD4 record. The total CD4 missing rate fall to 12.7%(Table 2). Ironically, after adding these missing CD4 counts, the CD4 count was still not significantly associate with TDR in univariable analysis (p=0.8), moreover it also did not meet the criterion for entering multivariable analysis (p<0.1).

Thirdly, more powerful methods were used to handle the missing data. The statistical method in this article stemmed for the “R in Action: Data Analysis and Graphics with R” by Robert I. Kabacoff. This book provided four method of handling miss data: list-wise deletion, case-wise deletion, simple imputation, and multiple imputation. Most popular statistic software used list-wise deletion to handle miss data by default. That was what had happened in the previous draft. As recommended in the textbook, we used multiple imputation(MI) in this revision and tried the other three methods. The results of the analysis using the four methods were consistent. Therefore, the missing of 12.7% CD4 count did not affect the final conclusion of this study.

Table 1. The CD4 missing by sampling period in the first draft

Period CD4 missing(n) Total(n) Rate of CD4 missing(%)

2001-2008 152 357 42.6

2009-2011 105 497 21.1

2012-2014 42 731 5.7

2015-2016 202 545 37.1

Total 501 2130 23.5

Table 2. The CD4 missing by sampling period in the new draft

Period CD4 missing(n) Total(n) Rate of CD4 missing(%)

2001-2008 143 357 40.1

2009-2011 97 497 19.5

2012-2014 16 731 2.2

2015-2016 16 545 2.9

Total 272 2130 12.7

Discussion

Line 337 – It is unclear how this is a “prospective” study if sequences prior to routine genotyping were examined for TDR retrospectively.

Our study is mainly a prospective study. In 2001-2005, it is retrospective, and in 2006-2016, it is prospective. Since 2,056 of 2,130 of the plasma were collected during 2006-2016, we believed that our study is 96.5%prospective.

Lines 389-392 – The discussion states that the data presented in this study “[assist] in the development of prevention and treatment strategies to influence the HIV epidemic within Beijing.” Further elaboration on how would be helpful.

We think our findings would be helpful in the following aspects: 

1. Because most patients are sensitive to first-line regimens, the routine genotype testing is not required, and there is no pressing need for expensive second-line regimens.

2. Although NFV is not used as part of first-line ART regimens in Beijing, we noted at least low-level TDR to these drugs in 1.9% of individuals. Thus, the use of NFV in Beijing should be examined with caution.

3. Vaccine designer in Beijing should take the fact that CRF01_AE, CRF07_BC and subtype B constituted more than 90% of all the clades into consideration in select appropriate candidate HIV strains. Viral load (VL) kit manufacturer should also know this in designing the primer to ensure accurate HIV RNA quantitation of non-B subtypes in Beijing.

Nonresponse should be discussed in further detail.

Yes, every HIV molecular epidemiology study will be inevitably subject to the caveat of non-response or the non-genotype. To our knowledge, there was no HIV molecular epidemiology with large sample size could genotype 100% of the samples. We believe that our 92% genotype rate is acceptable. In fact, non-genotype mainly took place for the plasma sample collected during period 2001-2008. It was supposed that the viral load tend to decline the longer they are preserved. We could not expect that the genotype rate in sample collected two decade before, preserved in refrigerator for 20 years, be as high as that for fresh blood collect today. In early days, preservation of sample was not seriously regarded. Samples were preserved wherever they could find, some times in -80℃ refrigerators, some times in -40℃, even in -20℃. Our laboratory then boasted only one -80℃ refrigerator. I could do nothing with this, because I was only a college student and I did not work here. We should thank our pioneer colleagues; it was they preserve the golden-like sample. As time passed, things changed a lot. We accumulate a lot of experience of genotyping during past decade. In the early stage, when in-house TDR genotype test was first introduced in Beijing. We copy the experiment without modification. We used subtype B primer, and we only concerned the genotype, without giving consideration to the non-genotype. What we got is what we analyzed. Then we could genotype 60%-80% of the samples. We did not take effort to repeat the test in the non-genotype. Things changed. Practice make perfect. During the past decade, I read carefully about 200 classical HIV epidemiology paper and I took more than ten HIV epidemiology course. In the end, we realize that the representation the non-genotype is of the same importance as the genotype. We also accumulated lots of practical experience of TDR test. We designed our own subtype specific primer. To improve the genotype rate, we usually carry out genotype test for three rounds. In first round, we use CRF01_AE specific primer. In second round, we use the CRF07_BC and subtype B specific primer both. In the third round, we use concentrated RNA template without adding water with all the three above mentioned primers. After three rounds of tests, we usually genotype about 90% of the samples. In this study, we compared the demographic characteristics of individuals with genotype vs. those with non-genotype. We show that the characteristics were broadly similar between individuals with and without genotyped virus. Therefore, we believed that the 8% non-response would not affect the final conclusion.

6. PLOS authors have the option to publish the peer review history of their article (what does this mean?). If published, this will include your full peer review and any attached files.

Do you want your identity to be public for this peer review? For information about this choice, including consent withdrawal, please see our Privacy Policy.

Reviewer #1: No

Reviewer #2: No

Yes, we visited the PACE web more than ten times; however, this URL was never available in Beijing, if only once. However, we fully understood the PLOS requirements for figures. That is TIFF or EPS in file; width: 789 – 2250 pixels (at 300 dpi), height maximum: 2625 pixels (at 300 dpi), in dimensions; 300 – 600 dpi for resolution; file Size<10 MB, Arial, Times, or Symbol font only in 8-12 point text within Figures. We are sure that our two figures met the requirement.

---

## [Decision Letter · Decision Letter 1]

2 Jan 2020

PONE-D-19-15622R1

Characterization of subtypes and transmitted drug resistance strains of HIV among Beijing residents between 2001-2016

PLOS ONE

Dear Dr. Lu,

Thank you for submitting your manuscript to PLOS ONE. After careful consideration, we feel that it has merit but does not fully meet PLOS ONE’s publication criteria as it currently stands. Therefore, we invite you to submit a revised version of the manuscript that addresses the points raised during the review process.

Please revise the manuscript based on the minor comments from Reviewer #2.  Also ensure that they manuscript has been reviewed thoroughly by a native English speaker and/or professional editing service.

We would appreciate receiving your revised manuscript by Feb 16 2020 11:59PM. To enhance the reproducibility of your results, we recommend that if applicable you deposit your laboratory protocols in protocols.io, where a protocol can be assigned its own identifier (DOI) such that it can be cited independently in the future. For instructions see: http://journals.plos.org/plosone/s/submission-guidelines#loc-laboratory-protocols

We look forward to receiving your revised manuscript.

Kind regards,

Jason Blackard, PhD

Academic Editor

PLOS ONE

Additional Editor Comments (if provided):

Please revise the manuscript based on the minor comments from Reviewer #2.  Also ensure that they manuscript has been reviewed thoroughly by a native English speaker and/or professional editing service.

Reviewers' comments:

Reviewer's Responses to Questions

**Comments to the Author**

1. If the authors have adequately addressed your comments raised in a previous round of review and you feel that this manuscript is now acceptable for publication, you may indicate that here to bypass the “Comments to the Author” section, enter your conflict of interest statement in the “Confidential to Editor” section, and submit your "Accept" recommendation.

Reviewer #1: All comments have been addressed

Reviewer #2: (No Response)

2. Is the manuscript technically sound, and do the data support the conclusions?

Reviewer #1: Yes

Reviewer #2: Partly

3. Has the statistical analysis been performed appropriately and rigorously? 

Reviewer #1: Yes

Reviewer #2: No

4. Have the authors made all data underlying the findings in their manuscript fully available?

Reviewer #1: Yes

Reviewer #2: No

5. Is the manuscript presented in an intelligible fashion and written in standard English?

Reviewer #1: Yes

Reviewer #2: Yes

6. Review Comments to the Author

Reviewer #1: The authors have addressed all the observations raised. The manuscript could be accepted for the publication.

Reviewer #2: The revised manuscript is substantially improved; however, one issue remains pertaining missing data.

The revised methods section states multiple imputation was used to handle missing data. However, the footnotes of the tables indicate differential sample sizes for certain variables. For instance, there were a total of 2130 sequences, but the number of sequences for each stratum of CD4 count interval in Table 3 only sums to 1858 (473+519+455+411). Further, the discussion states, "When missing, the CD4 counts was imputed, but imputation was only required for 454 12.8% of the study population." Did the authors only account for imputation in the regression analyses but not the descriptive data? If so, this is not standard practice and should be explained.

If the missing data were assumed to be missing at random and multiple imputation was truly employed to handle missing data, the methods section should provide more details on how and why multiple imputation was conducted. What variables had missing data and what percentage? What variables had missing data imputed? What variables were considered in the imputation model? How many imputed data sets were used?

7. PLOS authors have the option to publish the peer review history of their article (what does this mean?). If published, this will include your full peer review and any attached files.

Reviewer #1: No

Reviewer #2: No

---

## [Author Response · Author response to Decision Letter 1]

28 Jan 2020

Response to reviewers

PONE-D-19-15622R1

Characterization of subtypes and transmitted drug resistance strains of HIV among Beijing residents between 2001-2016

PLOS ONE

Dear Editor and Reviewers:

We would like to thank Plos One for giving use the opportunity to revise our manuscript. We thank the reviewer for this pertinent and insightful comment. We have carefully taken their comments into consideration in preparing our revision. Below is our response to their comments.

Reviewer #2: The revised manuscript is substantially improved; however, one issue remains pertaining missing data.

The revised methods section states multiple imputation was used to handle missing data. However, the footnotes of the tables indicate differential sample sizes for certain variables. For instance, there were a total of 2130 sequences, but the number of sequences for each stratum of CD4 count interval in Table 3 only sums to 1858 (473+519+455+411). Further, the discussion states, "When missing, the CD4 counts was imputed, but imputation was only required for 454 12.8% of the study population." Did the authors only account for imputation in the regression analyses but not the descriptive data? If so, this is not standard practice and should be explained.

Reply:

As discussed in this our previous response (see above), we used listwise deletion method to handle the missing data. We also attempted to use the more powerful method-multiple imputation. The introduction of multiple imputations added more problems in the analysis. As kindly mentioned by the reviewer, the imputation in the regression analysis is not a more conventional and widely used standard practice. This is now dropped in our final analysis. 

The data missing in our cohort mainly occurred for age (0.2%), transmission risk group (2.2%) and CD4 counts (12.8%). We think the effect of data missing in the first two variables on the association were negligible, therefore we mainly focus on the data missing in CD4 counts. Referring on the literature on how to handle the problem of miss data, at least ten HIV TDR articles published in Lancet infectious disease, Lancet HIV, CID, JID, AIDS, and Plos One within the past ten years were used to guide our analysis. Of the ten papers, only one in JID used listwise method to handle miss data[1]. The remaining nine articles did not mention what method was used to handle miss data[2-10]. Since the most popular statistic software used listwise deletion to handle miss data by default, we have assumed that the listwise method was used. On the contrary, multiple imputation was never mentioned at all.

Therefore, we decide to use listwise method rather the multiple imputation to handle the missing data throughout the article. Nevertheless, we did also try multiple imputation to handle CD4 missing in logistics analysis(S2 Table). The results from listwise and multiple imputation were consistent.

Moreover, when four sensitivity experiments were carried out by excluding individual sampling phase(S3-6 Table, the association between CD4 and TDR was still not significant.

If the missing data were assumed to be missing at random and multiple imputation was truly employed to handle missing data, the methods section should provide more details on how and why multiple imputation was conducted. What variables had missing data and what percentage? What variables had missing data imputed? What variables were considered in the imputation model? How many imputed data sets were used?

Reply: 

Age (0.2%), transmission risk group (2.2%) and CD4 counts (12.8%) had missing data. CD4 counts had missing data imputed. CD4 variable was considered in the imputation model. Five imputed data sets were used.

The program for multiple imputation in R was list below:

library(lattice) 

library(MASS)

library(nnet)

library(mice)

imp<-mice(data,m)

fit<-with(imp,analysis)

pooled<-pool(fit)

summary(pooled)

References

1 Wan-Lin Yang,Roger Kouyos,Alexandra U. Scherrer,et al.Assessing the paradox between transmitted and acquired HIV type 1 drug resistance mutations in the Swiss HIV cohort study from 1998 to 2012.J Infect Dis 2015;212:28-38.

2 L Marije Hofstra,Nicolas Sauvageot,Jan Albert,et al.Transmission of HIV drug resistance and the predicted effect on current first-line regimens in Europe.Clin Infect Dis 2016;62:655-663.

3 Santiago ávila-Ríos, Claudia García-Morales, Margarita Matías-Florentino,et al. Pretreatment HIV-drug resistance in Mexico and its impact on the effectiveness of first-line antiretroviral therapy:a nationally representative 2015 WHO survey,Lancet HIV 2016;3:e579-e591.

4 Theppharit Panichsillapakit,Davey M. Smith,Joe O. Wertheim,et al.Prevalence of transmitted HIV drug resistance among recently infected persons in San Diego, CA 1996-2013.J Acquir Immune Defic Syndr 2016;71:228-236.

5 Junko Hattori,Teiichiro Shiino,Hiroyuki Gatanaga, et al.Characteristics of transmitted drug-resistant HIV-1 in recently infected treatment-naive patients in Japan.J Acquir Immune Defic Syndr 2016;71:367-373.

6 Patrycja Machnowska, Karolin Meixenberger, Daniel Schmidt, et al. (2019) Prevalence andpersistence of transmitted drug resistancemutations in the German HIV-1 SeroconverterStudy Cohort. PLoS One2019; 14:e0209605.

7 Raph L. Hamers, Carole L. Wallis, Cissy Kityo,et al.HIV-1 drug resistance in antiretroviral-naive individuals in sub-Saharan Africa after roll-out of antiretroviral therapy:a multicentre observational study. Lancet Infect Dis 2011;11:750-759.

8 Donn J. Colby,Trevor A. Crowell,Sunee Sirivichayakul, et al.Declining trend in transmitted drug resistance detected in a prospective cohort study of acute HIV infection in Bangkok, Thailand.J Int AIDS Soc 2016;19:20966.

9 Junko Tanuma, Vo Minh Quang,Atsuko Hachiya, et al.Low prevalence of transmitted drug resistance of HIV-1 during 2008-2012 antiretroviral therapy scaling up in Southern Vietnam.J Acquir Immune Defic Syndr2014;66:358-364.

10 Vercauteren J , Wensing A , Vijver D V D , et al. Transmission of drug-resistant HIV-1 is stabilizing in Europe. Journal of Infectious Diseases, 2009, 200(10):1503-1508.

#2.  Also ensure that they manuscript has been reviewed thoroughly by a native English speaker and/or professional editing service.

Yes, we did.

We hope that our revision is acceptable, and we look forward to hearing from you soon.

---

## [Decision Letter · Decision Letter 2]

24 Feb 2020

PONE-D-19-15622R2

Characterization of subtypes and transmitted drug resistance strains of HIV among Beijing residents between 2001-2016

PLOS ONE

Dear Dr. Lu,

Thank you for submitting your manuscript to PLOS ONE. After careful consideration, we feel that it has merit but does not fully meet PLOS ONE’s publication criteria as it currently stands. Therefore, we invite you to submit a revised version of the manuscript that addresses the points raised during the review process.

Please make the minor updates requested by reviewer #2 prior to acceptance of your manuscript.

We would appreciate receiving your revised manuscript by Apr 09 2020 11:59PM. To enhance the reproducibility of your results, we recommend that if applicable you deposit your laboratory protocols in protocols.io, where a protocol can be assigned its own identifier (DOI) such that it can be cited independently in the future. For instructions see: http://journals.plos.org/plosone/s/submission-guidelines#loc-laboratory-protocols

We look forward to receiving your revised manuscript.

Kind regards,

Jason Blackard, PhD

Academic Editor

PLOS ONE

Additional Editor Comments (if provided):

Please make the minor updates requested by reviewer #2 prior to acceptance of your manuscript.

Reviewers' comments:

Reviewer's Responses to Questions

**Comments to the Author**

1. If the authors have adequately addressed your comments raised in a previous round of review and you feel that this manuscript is now acceptable for publication, you may indicate that here to bypass the “Comments to the Author” section, enter your conflict of interest statement in the “Confidential to Editor” section, and submit your "Accept" recommendation.

Reviewer #2: All comments have been addressed

2. Is the manuscript technically sound, and do the data support the conclusions?

Reviewer #2: Yes

3. Has the statistical analysis been performed appropriately and rigorously? 

Reviewer #2: Yes

4. Have the authors made all data underlying the findings in their manuscript fully available?

Reviewer #2: Yes

5. Is the manuscript presented in an intelligible fashion and written in standard English?

Reviewer #2: Yes

6. Review Comments to the Author

Reviewer #2: The authors have satisfied previous concerns. The revised manuscript now clearly states list-wise deletion was used to handle missing data in primary analyses. In addition, the manuscript now includes a sensitivity analysis using multiple imputation for the association of CD4 count and drug resistance (Supplemental Table 2).

Two minor comments:

Methods (page 9, Line 175): Please revise the sentence, "However when CD4 data were missing the multiple imputation was applied to perform logistics analysis." to read: "However, since 12.8% of data were missing for CD4 count, a sensitivity analysis was performed using multiple imputation to handle missing data (m=5)."

Please check p values in Table 3. The adjusted odds ratio for women was 0.26 (0.04-0.85) but the p value is >0.05 (i.e., p=0.06). The p value for this effect size and CL limit <1 would be expected to have a p value <0.05.

7. PLOS authors have the option to publish the peer review history of their article (what does this mean?). If published, this will include your full peer review and any attached files.

Reviewer #2: No

---

## [Author Response · Author response to Decision Letter 2]

6 Mar 2020

Dear Editor and Reviewers:

We would like to thank Plos One for giving use the opportunity to revise our manuscript. We thank the reviewer for this pertinent and insightful comment. We have carefully taken their comments into consideration in preparing our revision. Below is our response to their comments.

Two minor comments:

1. Methods (page 9, Line 175): Please revise the sentence, "However when CD4 data were missing the multiple imputation was applied to perform logistics analysis." to read: "However, since 12.8% of data were missing for CD4 count, a sensitivity analysis was performed using multiple imputation to handle missing data (m=5)."

Yes, we did it. This sentence is far more accurate than the former one.

2. Please check p values in Table 3. The adjusted odds ratio for women was 0.26 (0.04-0.85) but the p value is >0.05 (i.e., p=0.06). The p value for this effect size and CL limit <1 would be expected to have a p value <0.05.

 Thank you for your serious and careful proofreading. As the first author, though I have read the manuscript for more than 20 times, I still did not notice this inconsistence in Table 3. Honestly, when I first got this comment, my hair stood on end. Because this inconsistence could mean that either the OR or the p value is wrong. If it is true, the overall result of logistic analysis could be overthrew. That will be a nightmare for me. Fortunately, when I double checked the result of logistic analysis, I confirmed that there is no wrong in our logistic analysis. Therefore, the final question could be summed up as "Should the CI of OR always be consistent with the p value?" This question sounds like unprofessional. But it is the truth under our circumstance. Why is that so? We speculated that the sample size of women is too small to reach significance. The number of women in our study is trivial compared with the number of men( 151 vs 1979). Our recent national survey with larger sample size of women confirms that our speculation is correct. Moreover, the Chi-squared test also showed that the difference is non-significant(x-squared=3.0, p value=0.08).

Finally, we will like to express my appreciation for all these insightful comments and hard work in this reviewing. I learnt a lot of statistical knowledge in the submission. Yes, the process of submission is also the process of learning.

We hope that our revision is acceptable, and we look forward to hearing from you soon.

---

## [Editor Report · Decision Letter 3]

10 Mar 2020

Characterization of subtypes and transmitted drug resistance strains of HIV among Beijing residents between 2001-2016

PONE-D-19-15622R3

Dear Dr. Lu,

We are pleased to inform you that your manuscript has been judged scientifically suitable for publication and will be formally accepted for publication once it complies with all outstanding technical requirements.

With kind regards,

Jason Blackard, PhD

Academic Editor

PLOS ONE

Additional Editor Comments (optional):

None

Reviewers' comments:

None

---

## [Editor Report · Acceptance letter]

13 Mar 2020

PONE-D-19-15622R3 

Characterization of subtypes and transmitted drug resistance strains of HIV among Beijing residents between 2001-2016 

Dear Dr. Lu:

I am pleased to inform you that your manuscript has been deemed suitable for publication in PLOS ONE. Congratulations! Your manuscript is now with our production department. 

With kind regards,

on behalf of

Dr. Jason Blackard 

Academic Editor

PLOS ONE